# Investigation of the Membrane Localization and Interaction of Selected Flavonoids by NMR and FTIR Spectroscopy

**DOI:** 10.3390/ijms242015275

**Published:** 2023-10-17

**Authors:** Justyna Kapral-Piotrowska, Jakub W. Strawa, Katarzyna Jakimiuk, Adrian Wiater, Michał Tomczyk, Wiesław I. Gruszecki, Bożena Pawlikowska-Pawlęga

**Affiliations:** 1Department of Functional Anatomy and Cytobiology, Institute of Biological Sciences, Maria Curie-Sklodowska University, ul. Akademicka 19, 20-033 Lublin, Poland; justyna.kapral-piotrowska@mail.umcs.pl; 2Department of Pharmacognosy, Faculty of Pharmacy with the Division of Laboratory Medicine, Medical University of Białystok, ul. Mickiewicza 2a, 15-230 Białystok, Poland; jakub.strawa@umb.edu.pl (J.W.S.); katarzyna.jakimiuk@umb.edu.pl (K.J.); michal.tomczyk@umb.edu.pl (M.T.); 3Department of Industrial and Environmental Microbiology, Institute of Biological Sciences, Maria Curie-Sklodowska University, ul. Akademicka 19, 20-033 Lublin, Poland; adrian.wiater@mail.umcs.pl; 4Department of Biophysics, Institute of Physics, Maria Curie-Sklodowska University, ul. Pl. M. Curie-Sklodowskiej 1, 20-031 Lublin, Poland; wieslaw.gruszecki@mail.umcs.pl

**Keywords:** flavonoids, liposomes, NMR, FTIR, DPPC, EYPC

## Abstract

In this report, we discuss the effects of undescribed flavone derivatives, HZ4 and SP9, newly isolated from the aerial parts of *Hottonia palustris* L. and *Scleranthus perennis* L. on membranes. Interaction of flavonoids with lipid bilayers is important for medicinal applications. The experiments were performed with FTIR and NMR techniques on liposomes prepared from DPPC (dipalmitoylphosphatidylcholine) and EYPC (egg yolk phosphatidylcholine). The data showed that the examined polyphenols incorporate into the polar head group region of DPPC phospholipids at both 25 °C and 45 °C. At the lower temperature, a slight effect in the spectral region of the ester carbonyl group is observed. In contrast, at 45 °C, both compounds bring about the changes in the spectral regions attributed to antisymmetric and symmetric stretching vibrations of CH_2_ and CH_3_ moieties. Similarly, as in DPPC lipids, the tested compounds interact with the fingerprint region of the polar head groups of the EYPC lipids and cause its reorganization. The outcomes obtained by NMR analyses confirmed the localization of both flavonoids in the polar heads zone. Unraveled effects of HZ4 and SP9 in respect to lipid bilayers can partly determine their biological activities and are crucial for their usability in medicine as disease-preventing phytochemicals.

## 1. Introduction

Plants are natural resources for the discovery and development of novel drugs [1,2,3]. The health benefits and medicinal features of plants are attributed to the phytochemicals they contain, including flavonoids. Flavonoids are pigments that are mostly derived from benzo-γ-pyrone and are often incorporated into the human diet [4]. They exhibit a very large number of diverse properties, including antitumor, anti-inflammatory, analgesic, antimicrobial, antioxidant, and neuroprotective properties [5,6,7,8]. Because of these bioactivities, plants are natural sources of disease-preventing compounds with therapeutic potential. However, detailed knowledge about the effects of polyphenols is lacking. In the current study, we attempted to reveal the effects of two flavone derivatives named HZ4 and SP9 for the first time (Figure 1).

These derivatives were isolated from the aerial parts of *H. palustris* and *S. perennis*, and their interactions with and localization relative to membranes, as well as their influence on the dynamic and structural properties of model lipid membranes, were examined. In our previous work, we described four compounds obtained from the same plants [9]. These substances are especially interesting because no data have previously been obtained related to the bioactive potential of the phytochemicals isolated from the abovementioned plants coupled with membranes. Flavonoids must penetrate into or diffuse through lipid barriers to reach their site of action within cells. Inside membranes, different events occur that are associated with the lipid environment. Hence, interactions with membranes are among the important actions underlying the beneficial effects of flavonoids [4,5,6,10,11,12]. It has been shown that quercetin strongly decreases the microfluidity of the 1,2-dimyristoyl-sn-glycero-3-phosphocholine liposomal membrane bilayer at different depths and decreases the order of lipid molecule packing and increases hydration in the region of polar head groups. It also induces an increase in the zeta potential of the membrane and lowers the temperature and the enthalpy of the membrane phase transition. Such effects are associated with modification of membrane properties and result in prevention of oxidative stress [13]. In another study, it was revealed that the increase in liposomal membrane rigidity hinders the diffusion of free radicals and inhibits lipid peroxidation [14]. Moreover, membrane rigidification effects of flavonoids are a crucial factor for anti-cancer action of dietary flavonoids and can result in inhibition of the growth of cancer cells because of activation of membrane-mediated signaling pathways [15,16,17].

Currently, there are no published data about the mechanism of interaction of these new compounds (HZ4 and SP9) with the membranes. Therefore, the first aim of this work was to investigate the capability of HZ4 and SP9 to interact with unsaturated model membranes made of 1,2-diacyl-sn-glycero-3-phosphocholine from egg yolk (EYPC), resembling natural membranes, and with membranes made of dipalmitoylphospatidyl-choline (DPPC). Secondly, we decided to run FTIR measurements at two temperatures, namely, 25 °C and 45 °C, in order to examine structural effects in two essentially different thermotropic phases of lipid bilayer membranes formed with DPPC:Lβ’ (below the phase pretransition) and Lα above the main phase transition temperature. In such a way, we had access to information regarding the ordered and fluid phases. To achieve these goals, two techniques were applied. To determine the ability of polyphenols to incorporate, interact with, and change the structural and dynamic properties of model membranes, the ^1^H NMR method was applied [18,19]. The Fourier-transform infrared absorption spectroscopy (FTIR) technique was used in order to analyze molecular interactions between the tested molecules and membrane lipids. Hence, the present research is the first attempt to unravel the mechanisms of polyphenol derivatives’ action upon membranes. Alteration of membrane properties is crucial for natural drugs, such as flavonoids, and is coupled with production of multiple beneficial effects in cells. Additionally, using Chemaxon’s Protonation software, included in MarvinView 23.7.0 the selected physicochemical parameters of the investigated compounds were assessed to obtain additional knowledge about polyphenol–lipid interactions. These parameters are crucial for determining the affinity of natural compounds for lipid membranes.

## 2. Results

### 2.1. FTIR Investigation of DPPC Liposomes at 25 °C

At 25 °C, the membranes formed from DPPC were more rigid below the main phase transition temperature (41 °C). At this temperature, HZ4 affected different regions of the membranes in various ways. The examined flavonoid caused reconstruction of the water layers (Figure 2a).

In the spectral region attributed to symmetric and antisymmetric stretching vibrations of CH_2_ and CH_3_ moieties, HZ4 did not cause any changes. Hence, HZ4 did not alter the fluidity of the membrane at this temperature (Figure 2b). Shallow incorporation of HZ4 into the membrane was evidenced by the slight effect on the ester carbonyl groups at the examined temperature, as a shift of the related spectral peaks toward lower frequencies (1723 cm^−1^) was observed (Figure 2c). The spectral region corresponding to the choline group was not affected by HZ4 (Figure 2e). Spectral effects resulting from the presence of HZ4 in the DPPC membranes indicated the localization of the investigated flavonoid in the polar head group region. HZ4 was incorporated into the region of the phosphate groups and led to their reorganization. In the spectral region attributed to antisymmetric stretching vibrations of the PO_2_^-^ groups, HZ4 caused a shift of the observed peaks toward lower frequencies and decreased the intensity of these bands. HZ4 increased the intensity of the bands corresponding to C–O–P–O–C vibrations and simultaneously caused a shift of the spectral peaks toward lower frequencies (1051 cm^−1^) (Figure 2d).

As shown in Figure 3, SP9 was incorporated into the polar head group region of the phospholipids and slightly affected the ester carbonyl groups of DPPC, resulting in a shift of the corresponding spectral peaks toward lower frequencies (1733 cm^−1^) (Figure 3c). A slight effect on the choline groups was also observed, as the oscillator strength of the associated spectral peaks increased (Figure 3e). Similarly to HZ4, this compound caused reorganization in the water layer of the membranes (Figure 3a). Spectra showing the region corresponding to the antisymmetric and symmetric stretching vibrations of the CH_2_ and CH_3_ groups are presented in Figure 3B. It can be deduced that the examined compound did not alter the membrane fluidity (Figure 3b). SP9 strongly affected the spectral region associated with symmetric stretching vibrations of the phosphate groups and decreased the oscillator strength of these peaks. This effect was explained by the cleavage of hydrogen bonds. In the region corresponding to C–O–P–O–C vibrations, a shift of the observed peaks toward lower frequencies was observed. A positive band centered at 1035 cm^−1^ was visible in the difference spectrum (Figure 3d).

### 2.2. FTIR Experiments on DPPC Membranes at 45 °C

Large changes were observed in the spectral region corresponding to the O–H stretching vibrations (Figure 4a).

A positive band appeared in the difference spectrum at 3323 cm^−1^. In the region corresponding to acyl chains, positive bands appeared in the region associated with antisymmetric and symmetric stretching vibrations of CH_3_ groups (2969 cm^−1^, 2882 cm^−1^) (Figure 4b). HZ4 did not affect the carbonyl ester groups (Figure 4c). A small effect on the phosphate groups was observed. The examined compound caused some reorganization of the abovementioned region of the membrane (Figure 4d). A very strong influence was observed in the spectral region attributed to antisymmetric N+–CH_3_ stretching vibrations (choline groups). There was a shift toward lower frequencies coupled with a decrease in the oscillator strength (Figure 4e).

Similarly to HZ4, SP9 strongly affected the spectral region corresponding to O–H stretching modes in water molecules associated with the membranes via hydrogen bonds. SP9 caused reorganization of the corresponding membrane region (Figure 5a).

Another spectral effect (slight effect) concomitant with the addition of SP9 into the membranes was visible in the region attributed to the antisymmetric and symmetric stretching vibrations of CH_2_ and CH_3_ groups. There was a visible shift toward higher frequencies (2936 cm^−1^) that indicated fluidization of the membranes (Figure 5b). Little impact on the ester carbonyl group was observed (Figure 5c). The most pronounced effect was observed in the fingerprint region of the polar head groups, especially in the spectral region corresponding to the polar head group vibrations, which were observed as antisymmetric stretching of the PO_2-_ groups (1247 cm^−1^) and symmetric stretching of the PO_2-_ groups (1089 cm^−1^). Thus, SP9 caused the polar head group region to reorganize by intercalating into the membranes (Figure 5d). No significant effect was observed in the spectral region associated with the choline group (Figure 5e).

### 2.3. FTIR Investigation of the EYPC Membranes

The most pronounced spectral effect accompanying the presence of HZ4 in the membranes was observed in the spectral regions corresponding to the symmetric stretching vibrations of the PO_2-_ groups (Figure 6d). The increase in the oscillator strength and the slight shift of the peaks toward lower frequencies (positive band in the difference spectrum at 1086 cm^−1^) indicated the localization of the examined flavonoid. HZ4 did not affect the region corresponding to the O–H stretching vibrations, the ester carbonyl groups, or the choline group (Figure 6a,c,e).

In contrast, SP9 had a more significant effect on the membranes. First, a significant impact on the spectral region corresponding to the O–H stretching vibrations was observed (Figure 7a).

SP9 increased the fraction of water bound to the membranes. Additionally, in the spectral region corresponding to the acyl chains of the lipids (in the region attributed to the symmetric and antisymmetric stretching vibrations of CH_2_ and CH_3_ moieties), SP9 caused a shift of the observed peaks toward higher frequencies (2934 cm^−1^) (Figure 7b). This effect indicated that the lipid membranes fluidized. Similarly to HZ4, SP9 did not affect the ester carbonyl group; however, unlike HZ4, SP9 slightly affected the choline group (Figure 7c,e). The phosphate group region was reorganized under the influence of SP9 (Figure 7d). Positive bands centered at 1291 and 1030 cm^−1^ were visible in the difference spectrum

### 2.4. NMR Experiments on the DPPC Liposomes

The ^1^H NMR spectra of pure DPPC liposomes and DPPC liposomes doped with HZ4 are presented in Figure 8.

The addition of HZ4 resulted in a decrease (by 21%) in the splitting parameter of the resonance maximum corresponding to the polar head group (δ). This change indicated that little ordering occurred in the polar head group region of the membrane. Moreover, the examined flavonoids induced the formation of multilamellar liposomes, as revealed by the increased I_out_/I_in_ ratio from 0.42 in pure DPPC to 0.62 in liposomes with addition of HZ4 and 0.68 in liposomes with addition of SP9. The full width at half height (ν) value of ^1^H NMR for the CH_2_ groups decreased (by 40%), which indicated a strong fluidizing effect in the hydrophobic region. A certain effect on the polar head group region of the membrane was observed in the liposomes upon SP9 addition (Figure 9). A decrease in ν by 21% (inner leaflet of membrane) was observed.

Almost no changes were observed in the ν parameter in the splitting parameter of the resonance maximum (δ) corresponding to the polar head group region. In turn, similarly to the results observed with HZ4, a decrease (by 23%) in the full width at half height (ν) value corresponding to the CH_2_ groups was noted.

### 2.5. Physicochemical Parameters of the Studied Compounds HZ4 and SP9

In polyphenol–lipid interactions, an affinity for lipid membranes is essential. Thus, we predicted some of the physicochemical parameters of our compounds. These parameters are shown in Table 1.

One of the calculated parameters was the logarithm of the partition coefficient, logP, which describes the lipophilicity or hydrophobicity of a compound. This property is associated with the ability of polyphenols to incorporate into membranes as well as more effective penetration of compounds into membranes [18,20]. The other parameter was the acid dissociation constant, pKa, which characterizes the state in which a compound is in equilibrium between its undissociated and dissociated forms [21]. The last parameter considered was the logarithm of the distribution coefficient, logD. This value is a ratio of the sum of the concentrations of the ionized and nonionized forms of the compounds in two solvent phases, one of which is always aqueous [22]. The value is dependent on the pH of the aqueous phase for the ionized form. For a summary of the parameters, see Table 1.

## 3. Discussion

Thus, in this work, the molecular interactions of two flavone derivatives (HZ4 and SP9) isolated from the aerial parts of *H. palustris* and *S. perennis* with liposomes composed of dipalmitoylphosphatidylcholine (DPPC) and egg yolk phosphatidylcholine (EYPC) were investigated. In this work, Fourier-transform infrared (FTIR) spectroscopy was used, and the bands corresponding to vibrations in various lipid-related regions of the spectra of DPPC membranes at 25 and 45 °C were analyzed.

At 25 °C, which is below the main phase transition temperature (41 °C), the membranes composed of DPPC were more rigid. The analyses showed that the examined polyphenols were incorporated into the membrane region containing the polar head group of the DPPC phospholipids at this temperature. The main spectral regions affected by the investigated polyphenols were the regions attributed to antisymmetric stretching vibrations of the PO_2_- groups (HZ4) (a shift of the observed peak toward lower frequencies was observed, and the intensity of this peak decreased), the region corresponding to symmetric stretching vibrations of the phosphate groups (SP9), and the bands corresponding to C–O–P–O–C vibrations (increased peak intensity and a shift toward lower frequencies) (both HZ4 and SP9) and choline groups (SP9). Simultaneously, shallow incorporation of these compounds within the membranes was indicated by the slight effect on the spectral region attributed to the ester carbonyl group. Neither compound affected the acyl chains, but both caused the water layer of the membrane to reorganize. Similar to the results at 25 °C, at 45 °C, HZ4 and SP9 induced changes in the spectral region corresponding to the O–H stretching modes in the water molecules associated with the membranes via hydrogen bonding. These changes were attributed to reorganization of this membrane region. The same slight effect was also observed in the ester carbonyl group region in the spectrum of the membrane integrated with SP9 (HZ4 had no effect). In contrast, at 45 °C, both compounds induced some spectral changes in the region corresponding to the antisymmetric and symmetric stretching vibrations of the CH_2_ and CH_3_ groups. The presence of SP9 in the membranes resulted in a slight spectral effect, namely, a shift in the observed peaks toward higher frequencies (2936 cm^−1^), indicating fluidization of the membranes. In contrast, HZ4 resulted in the appearance of positive bands in the region of the antisymmetric and symmetric stretching vibrations of CH_3_ groups. The most pronounced effect was observed in the fingerprint region of the polar head groups, as the examined flavonoids caused reorganization of this area after intercalation. For HZ4, in the spectral region corresponding to antisymmetric N^+^–CH_3_ stretching vibrations (choline groups), a shift of the observed peaks toward lower frequencies and a decrease in the oscillator strength were observed, while, for SP9, the peaks attributed to antisymmetric stretching of the PO_2-_ groups and symmetric stretching of the PO_2_- groups were affected.

The results of our investigations demonstrated that the examined flavonoid derivatives interacted mainly with the polar region of the liposomes. As shown by our FTIR study on DPPC liposomes, the spectral effects caused by the two tested compounds were mainly changes in peak intensity. In addition, spectral shifts of the bands corresponding to vibrations of the C–O–P–O–C moiety, the antisymmetric and symmetric stretching vibrations of the PO_2_- groups, or the antisymmetric N^+^–CH_3_ stretching vibrations toward lower frequencies were observed. The incorporation of the flavonoids into the polar head region of the liposomes has also been demonstrated and supported by many authors [11,23,24,25,26,27,28]. It has been claimed that the lipid glycerol segment and the region containing acyl chains adjacent to the choline group are the lipid membrane regions most affected by polyphenols [20]. This location and distribution agree well with our findings. For both examined compounds, we observed that the spectral band attributed to the C–O–P–O–C group was sensitive to the presence of HZ4 and SP9 in the membranes. A shift of this peak toward lower frequencies was concomitant with an increase in the oscillator strength. These changes can be explained by the formation of hydrogen bonds between the head groups of DPPC, indicating that the flavonoids were incorporated into DPPC via hydrogen bonding in this region, thus revealing the localization of the flavonoids. Our results are consistent with those of other studies in which FTIR was applied to investigate DPPC liposomes and in which the authors examined flavonoid interactions with artificial membranes [28,29]. At the surface of bilayers, these interactions can determine the ability of active polyphenols to intercalate into lipids and exert antioxidant activity against radicals [5,11].

In the current FTIR investigation, a shift of spectral peaks toward higher frequencies was observed in the region corresponding to DPPC acyl chains for liposomes examined at a temperature of 45 °C, which is above the main phase transition temperature. Hence, we concluded that both polyphenols weakened the interactions between the acyl chains, and, in particular, SP9 induced fluidization in the hydrophobic region of the membrane. The effects of flavonoids on artificial and natural membranes often involve alterations in membrane fluidity and the packing order of lipids [4,9,13,30]. Once incorporated into the membrane, flavonoids can cause different changes. The results obtained for the DMPC membranes showed that the temperature and enthalpy of the membrane transition were decreased by quercetin incorporation [13]. Similarly, other researchers observed a pronounced increase in ultrasound absorption combined with a shift in the phase transition temperature toward lower values after quercetin was incorporated into the DPPC membranes [31]. Increased membrane fluidity and alterations in acyl chain conformations were also documented in other studies [28,32].

Structural properties of chemical compounds, among them, flavonoids, determine the ability to penetrate the lipid bilayers. Similarly, the presence of some additional substituents such as hydroxyl groups, methoxy groups, and sugar moieties, together with their distribution in the molecule, can affect the membrane interactivity of flavonoids [18,33,34]. Our examined polyphenol derivatives have some structural traits. A characteristic feature of SP9 is the presence of sugars molecules. Both the examined compounds have hydroxyl groups linked with carbon C-5 (HZ4, SP9) and carbon C-7 (SP9). Additionally, both were found to possess methoxy groups joined to carbon C-2′ and C-6′ (HZ4) and to carbon C-3′ (SP9). At the same time, in the current study, both examined compounds incorporated into lipid membranes. It is known that hydrophobicity makes the polyphenols penetrate more deeply into membranes [9,15]. Hence, we assessed some from the physicochemical parameters of our polyphenols. Compound HZ4 demonstrated higher logP and logD values (2.18 and 1.9, respectively) than compound SP9, indicating that it was more lipophilic with similar pKa values (6.29 and 5.02, respectively).

In this study, we also employed membranes prepared from EYPC. In these liposomes, double bonds in fatty acids prevented the association of adjacent acyl chains. Compared to the membranes formed from DPPC, these membranes were characterized by a less compact structure. The data obtained in this work revealed that both examined compounds intercalated into the membranes within the polar head region. The most pronounced spectral effect accompanying the presence of both compounds in the membranes was observed in the spectral regions corresponding to the phosphate groups. HZ4 triggered an increase in the oscillator strength of peaks in these regions and a slight shift of these peaks toward lower frequencies, whereas SP9 resulted in reorganization of the corresponding membrane region. Our FTIR data obtained for the DPPC liposomes also showed that the flavonoids were incorporated into the membranes in this region. Moreover, these results are consistent with those in our previous work in which the interaction of other polyphenols with artificial bilayers was investigated [9]. Additionally, our results are consistent with those of a variety of past studies [11,25,26,28,30,35]. The head group region was found to be the main and preferential location of flavonoids within the lipid bilayer. The result of SP9 inclusion in the EYPC membranes was a spectral shift of peaks in the region attributed to acyl chains toward higher frequencies, implying that the flavonoid molecule induced fluidization of the hydrophobic core of the lipid membranes. The same effect was observed for the DPPC liposomes.

The findings obtained by applying FTIR correspond well with the NMR data obtained in the current study. The polar head group region was determined to be the site of intercalation for the examined compounds. A small decrease in the splitting parameter of the resonance maximum corresponding to polar head groups (δ) was observed. These changes indicate that little ordering occurred in the polar head group region of the membranes. Similarly, a decrease in the motional freedom of the polar head group region was noted in another study. The authors investigated the effect of genistein on the polar head group region. Analyses of proton resonance in the choline group of pure membranes and membranes with the addition of a specific isoflavone were performed [29]. In another study, the ordering effect of quercetin on the head group region of liposomes was also observed. Additionally, flavonol induced structural changes in the cholesterol/sphingolipid-enriched domains [28,36]. Changes were observed in the *ν* parameter corresponding to the hydrophobic region of the DPPC liposomes after the two polyphenols of interest were integrated. HZ4 and SP9 caused a decrease in the full width at half height (ν) value corresponding to the CH_2_ groups. The results obtained by other authors are consistent with our data. Increased motional freedom of alkyl chains, as manifested by narrowing of the spectral peaks attributed to the CH_2_ and terminal CH_3_ groups of the lipid alkyl chains, in quercetin-containing samples of liposomes was observed through NMR [28]. Flavonoids can act at different depths in membranes and cause the lipid packing order to change [32].

## 4. Materials and Methods

### 4.1. Chemicals

The examined HZ4 (5-hydroxy-2′,6′-dimethoxyflavone) and SP9 (5,7-dihydroxy-3′-methoxy-4′-acetoxyflavone-8-*C*-*β*-D-xyloside-2′′-*O*-(4′′′-acetoxy)-glucoside) compounds applied in these studies were dissolved in ethanol (EtOH) (Merck, Germany). The aboveground parts of *Hottonia palustris* L. (Primulaceae) and *Scleranthus perennis* L. (Caryophyllaceae) were the sources of these secondary metabolites (HZ4 and SP9). Through a multistep isolation process, the products were obtained in 98% purity by HPLC [37,38]. 1,2-Dipalmitoyl-sn-glycero-3-phosphocholine (DPPC) and egg yolk lecithin (EYPC) were purchased from Sigma Chemical Co. (St. Louis, MO, USA). Deuterium oxide (D_2_O) was purchased from ARMAR Chemicals Co. (Döttingen, Switzerland).

### 4.2. Preparation of Liposomes

Shaking was performed to create multilamellar liposomes for FTIR measurements [28]. Saturated DPPC liposomes, which resulted in an ordered phase that was interrupted by the incorporation of various flavonoids, were used in this study. The concentration of the lipids was 2.72 × 10^−2^ M. The concentration of flavonoids was 5 mol% with respect to the lipids. For the EYPC liposomes, the concentration of lipids was 2.64 × 10^−2^ M. The dispersion of multilamellar DPPC and EYPC liposomes was prepared by mixing solutions of the respective compounds, followed by evaporation of the solvent first in a stream of nitrogen and subsequently under a vacuum (4 h). The samples were then hydrated with a water/D_2_O mixture (95:5, *v*:*v*) and vigorously shaken at a temperature above the main phase transition temperature of the lipid (41 °C) to achieve optical homogeneity. For the NMR measurements, sonication was conducted. Briefly, flavonoids and phospholipids were co-dissolved in an ethanol/chloroform mixture at a particular concentration [29]. The concentration of the lipids in the sample was 3.2 × 10^−2^ M, and that of the flavonoids was 3.2 × 10^−4^ M. Then, the samples were evaporated under a stream of nitrogen and left under vacuum overnight to remove residual solvents. Subsequently, after hydration with D_2_O, the specimens were vigorously agitated (1 h) on a shaker at a temperature above the main phase transition temperature of the lipids (41 °C). Then, the lipid dispersion was sonicated (8 × 3 s, power: 20 kHz) at 4 °C (in a water–ice bath) with a VCX sonicator (Sonics Vibra Cell™, Newtown, CT, USA) to yield a homogeneous lipid suspension.

### 4.3. Nuclear Magnetic Resonance (^1^H NMR) Spectroscopy of DPPC Liposomes

Shortly before the NMR experiments, 4 mM PrCl_3_ was added to the lipid–flavonoid suspension. ^1^H NMR spectra were acquired on an FT NMR Bruker AVANCE 500 NMR spectrometer (Ettingen, Germany) using a 5 mm probe in deuterated water at 60 °C, with pulsed field gradient capabilities at a spinning speed of 500 MHz. The other ^1^H NMR parameters were as follows: spectral window, 10,333 Hz; digital resolution, 0.1576 Hz; pulse width, 6.5 μs; and acquisition and delay times, 3.17 s and 1 s, respectively. The number of scans was 32.

### 4.4. Fourier-Transform Infrared (FTIR) Spectroscopy

The infrared absorption spectra of polyphenols, pure liposomes (DPPC and EYPC), and liposomes with doped flavonoids were recorded using a Fourier-transform infrared absorption spectrometer equipped with the attenuated total reflection configuration (ATR-FTIR) [28]. Through evaporation from water and D_2_O, the samples were deposited on a crystal. Then, the spectra were recorded with a Nicolet iS50R from Thermo Scientific (Waltham, MA, USA). An internal reflection element (a diamond prism) was used as the attenuated total reflection element (ATR). Ten scans were performed for each sample, and then Fourier transformation was performed. Then, the results were averaged for each measurement. Absorption spectra at a resolution of one data point every 2 cm^−1^ were collected in the region between 4000 and 400 cm^−1^ using a clean crystal as the background. Purging was performed with argon gas for 40 min before measurements and then constantly during measurements. The experiments were performed at 25 °C for EYPC liposomes and at 25 and 45 °C for DPPC liposomes. Spectral analysis was performed with Grams AI software version 9.1 from ThermoGalactic (Waltham, MA, USA).

### 4.5. Calculation of Physicochemical Parameters

The log P, pKa, and logD values for the examined compounds were predicted using Chemaxon’s Protonation software included in MarvinView 23.7.0 (Basel, Switzerland).

## 5. Conclusions

In this paper, for the first time, the interactions of two flavone derivatives HZ4 and SP9 with model membranes were studied. We pointed out the localization of these polyphenols within membranes as well as their impact on the dynamic and structural properties of liposomes membranes. Both polyphenols weakened the interactions between the acyl chains and induced fluidization in the hydrophobic region of the membrane. The data also showed that the examined polyphenols were incorporated via hydrogen bonding into the membrane region containing the polar head group of phospholipids at both 25 °C and 45 °C. At the lower temperature, shallow incorporation was found as it was assessed on the basis of a slight effect on the spectral region attributed to the ester carbonyl groups. In contrast, at 45 °C, both compounds gave rise to slight effects in the spectral regions attributed to antisymmetric and symmetric stretching vibrations of CH_2_ and CH_3_ moieties, indicating deeper incorporation. This temperature-dependent effect may partly determine biological activity of both tested compounds and is critical for the usability of these compounds.

## Figures and Tables

**Figure 1 ijms-24-15275-f001:**
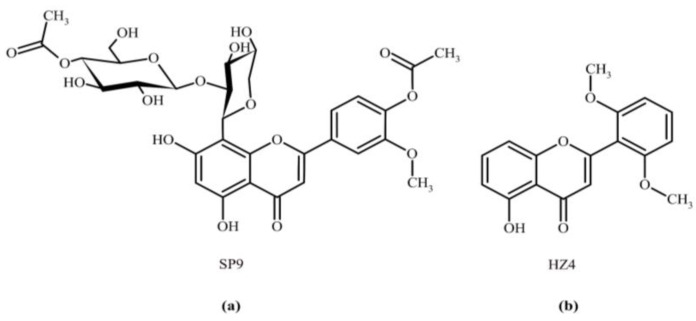
Chemical structures of (**a**) HZ4 and (**b**) SP9.

**Figure 2 ijms-24-15275-f002:**
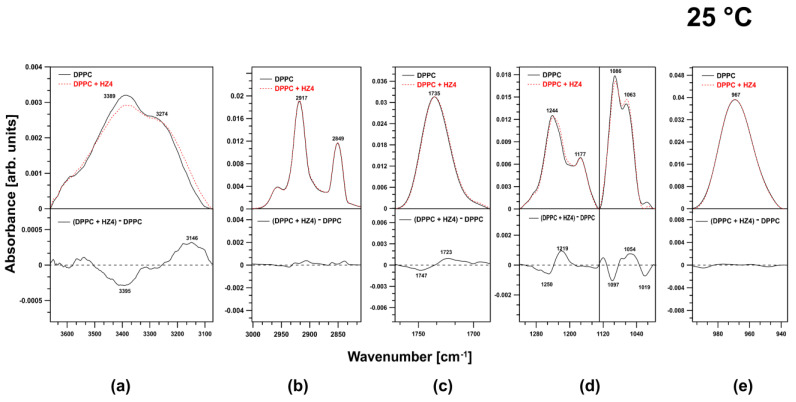
Characteristic region of the ATR-FTIR spectra of liposomes formed with pure DPPC (continuous line) and DPPC upon the addition of HZ4 (dashed line) and difference spectra at 25 °C. The spectra were normalized by dividing by the area beneath the bands in the regions of (**a**) 3556–3140 cm^−1^, (**b**) 3001–2747 cm^−1^, (**c**) 1800–1672 cm^−1^, (**d**) 1320–996 cm^−1^, and (**e**) 996–934 cm^−1^.

**Figure 3 ijms-24-15275-f003:**
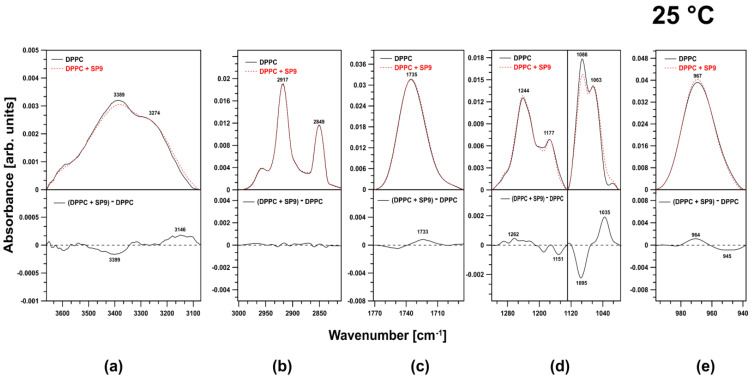
Comparison of FTIR spectra in the regions of (**a**) 3556–3140 cm^−1^, (**b**) 3001–2747 cm^−1^, (**c**) 1800–1672 cm^−1^, (**d**) 1320–996 cm^−1^, and (**e**) 996–934 cm^−1^. The graph shows spectra of liposomes created with pure DPPC (continuous line) and liposomes with the addition of SP9 (dashed line) at 25 °C. The lower panels show the difference spectra.

**Figure 4 ijms-24-15275-f004:**
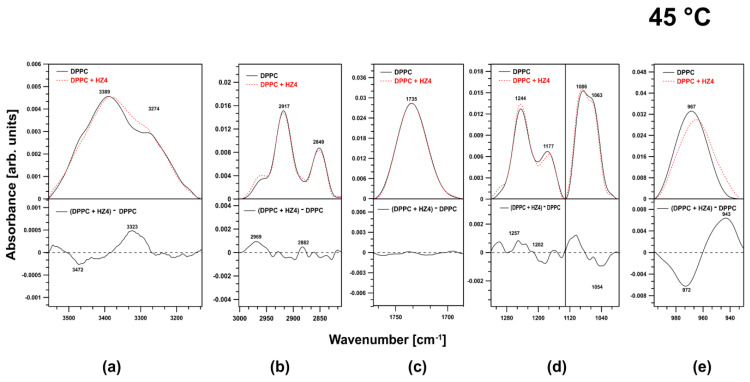
ATR-FTIR spectra of DPPC vesicles (continuous line) and vesicles with the addition of HZ4 (dashed line) and difference spectra (lower parts of the graph) in the regions of (**a**) 3556–3140 cm^−1^, (**b**) 3001–2747 cm^−1^, (**c**) 1800–1672 cm^−1^, (**d**) 1320–996 cm^−1^, and (**e**) 996–934 cm^−1^ at 45 °C.

**Figure 5 ijms-24-15275-f005:**
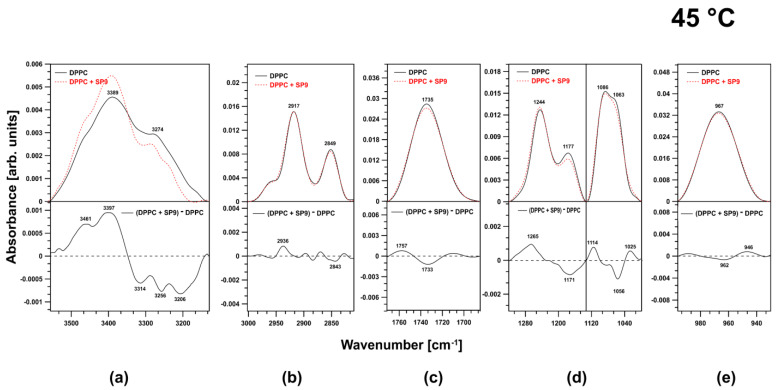
Typical ATR-FTIR spectral regions from (**a**) 3556–3140 cm^−1^, (**b**) 3001–2747 cm^−1^, (**c**) 1800–1672 cm^−1^, (**d**) 1320–996 cm^−1^, and (**e**) 996–934 cm^−1^ in the spectra of liposomes created with pure DPPC (continuous line) and liposomes treated with SP9 (dashed line) and the difference spectra (lower panels) at 45 °C.

**Figure 6 ijms-24-15275-f006:**
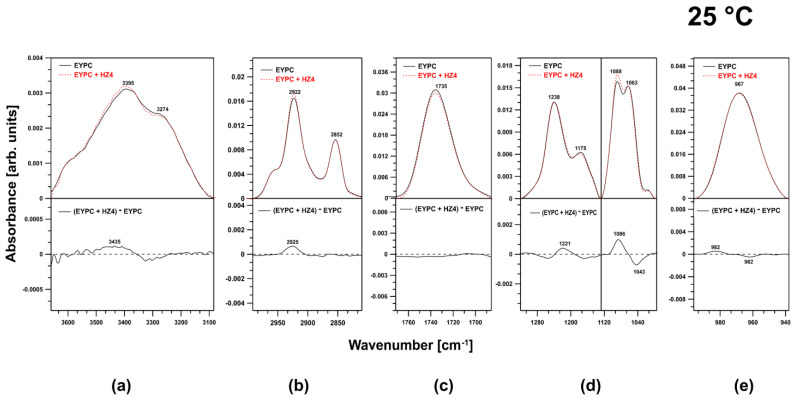
Representative infrared spectra of EYPC liposomes (continuous line) and liposomes plus HZ4 (dashed line). The difference spectra are shown in the bottom panels. The spectra were normalized by the area below the bands in the regions of (**a**) 3556–3140 cm^−1^, (**b**) 3001–2747 cm^−1^, (**c**) 1800–1672 cm^−1^, (**d**) 1320–996 cm^−1^, and (**e**) 996–934 cm^−1^.

**Figure 7 ijms-24-15275-f007:**
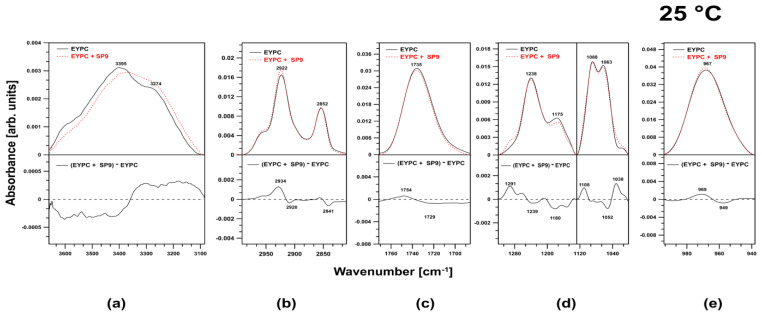
The figure presents the effect of SP9 on the characteristic regions (**a**) 356–3140 cm^−1^, (**b**) 3001–2747 cm^−1^, (**c**) 1800–1672 cm^−1^, (**d**) 1320–996 cm^−1^, and (**e**) 996–934 cm^−1^ in the spectra of EYPC liposomes. In the lower part of the graph, difference spectra are shown. The spectra were normalized by dividing by the area under the curve in each region.

**Figure 8 ijms-24-15275-f008:**
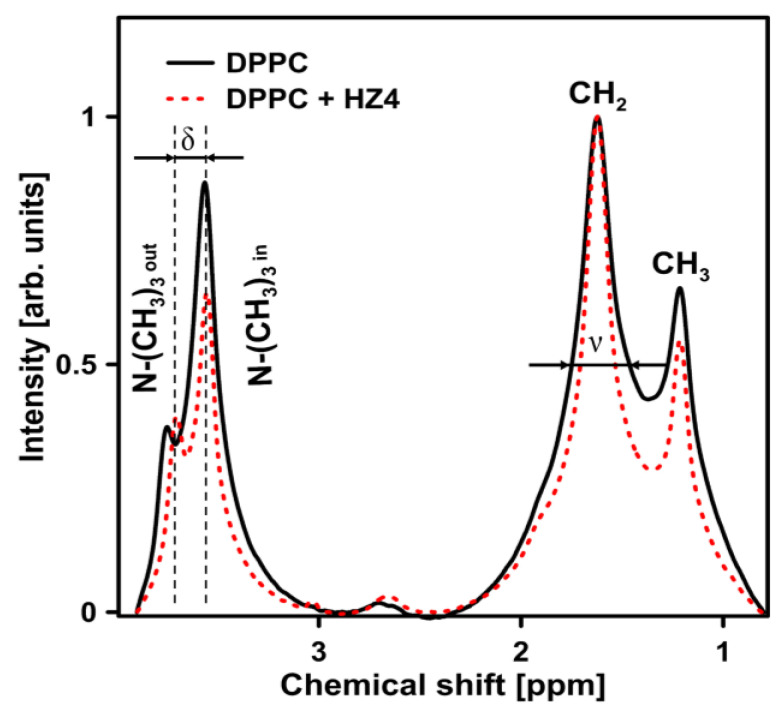
^1^H NMR analysis of liposomes formed with pure DPPC (continuous line) and DPPC with HZ4 (dashed line). To split the ^1^H NMR peak attributed to −N+CH_3_, PrCl_3_ was added to the samples before the experiments were performed. The resonance line assignment and parameters used in the spectral analysis are shown in the graph. The following parameters were determined: the full width at half height (ν) and the splitting parameters of the resonance maximum corresponding to polar head groups (δ).

**Figure 9 ijms-24-15275-f009:**
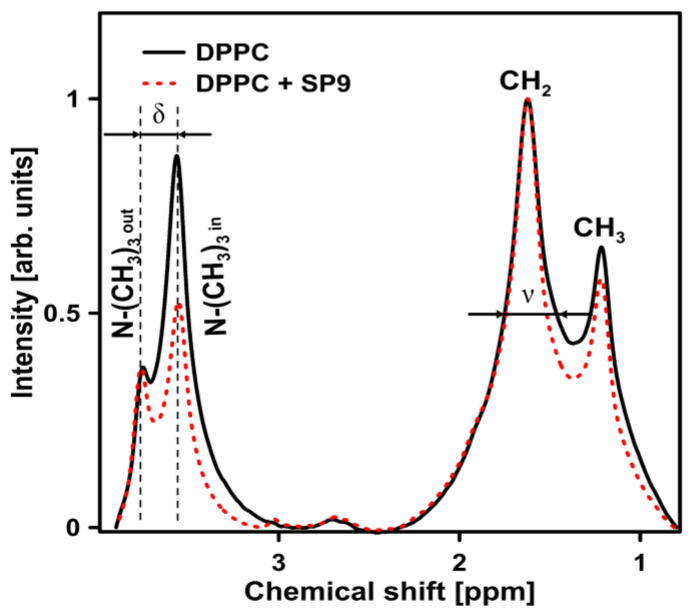
^1^H NMR spectra showing the effect of a new compound isolated from S. perennis (SP9) on DPPC liposomes. Analyzed parameters such as the full width at half height (ν) and splitting parameters of the resonance maximum corresponding to the polar head groups (δ) are presented in the graph. The ^1^H NMR peak attributed to −N+CH_3_ was split by adding PrCl_3_ before the experiments were conducted.

**Table 1 ijms-24-15275-t001:** Physicochemical parameters of HZ4 and SP9 evaluated by Chemaxon’s Protonation software.

ID	Name	MW (g/mol)	LogP	pKa Acidic	LogD (pH 4)
HZ4	5-hydroxy-2′,6′-dimethoxyflavone (C_17_H_14_O_5_)	298.294	2.18	6.29	1.9
SP9	8-C-*β*-D-xyloside-(2′′-*O*-(4′′′-acetoxy)glucoso)-5,7-dihydroxy-3′-methoxy-4′-acetoksyflawone (C_31_H_34_O_17_)	678.596	−2.42	5.02	−3.5

## Data Availability

The data presented in this study are available on request from corresponding author.

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
