# Peer review of "Investigation of the Membrane Localization and Interaction of Selected Flavonoids by NMR and FTIR Spectroscopy"

_ijms, 2023, doi:10.3390/ijms242015275_

Round 1

Reviewer 1 Report

Hi Authors,

The article titled 'Investigation of the membrane localization and interaction of 2 selected flavonoids by NMR and FTIR spectroscopy' is written well and experiments conducted in this study are appropriate. Please find my comments below.

1.  Abstract - Expansion of acronyms such as DPPC and EYPC is encouraged as it can help the readers to understand easily. 

2. Abstract - the sentences can be paraphrased so that it is clear and simple.

3. Introduction - can be improved.

4. Why two temperature conditions (25 and 45 degrees) were used in FTIR experiments?

5. Results and discussion - written well.

6. Conclusion part can be improved.

Minor revisions is needed.

Author Response

Reviewers 1  comments:

Thank you very much for your valuable  commnets . We have improved our manuscript according to these  commnets.

  1. Abstract - Expansion of acronyms such as DPPC and EYPC is encouraged as it can help the readers to understand easily. 

Now in abstract both acronyms are present..

  1. Abstract - the sentences can be paraphrased so that it is clear and simple.

The whole abstract was paraphrased  and now is incorporated into the revised version of the manuscript.

  1. Introduction - can be improved.

Now Introduction was improved and it is incorporated into the revised version of the manuscript.

  1. Why two temperature conditions (25 and 45 degrees) were used in FTIR experiments?

We decided to run FTIR measurements in two temperatures, namely 25 oC and 45 oC, in order to examine structural effects in two essentially different thermotropic phases of lipid bilayer membranes formed with DPPC: Lβ (below the phase pretransition) and Lα above the main phase transition temperature. In such a way we have access to information regarding the ordered and fluid phases. Such explanation was also mentioned in the body of the main text.

  1. Results and discussion - written well.
  2. Conclusion part can be improved.

Conclusion part was improved and now is incorporated into the revised version of the manuscript.

Reviewer 2 Report

The work is the authors established mechanisms of action for unfamiliar flavonoids 410 isolated from the aerial parts of H. palustris and S. perennis and examined their interaction and localization within lipid membranes. The manuscript is well explained the obtained data. My suggestion is to improve the introduction. The aim and novelty of the work are missing. It is better for the readers to discuss the output in detail, compared to existing knowledge..

The English language is fine.

Author Response

Reviewer 2 comment:

Thank you very much for this comment. We considered this comment and now manuscript is changed.

My suggestion is to improve the introduction. The aim and novelty of the work are missing.

Now Introduction was improved and it is incorporated into the revised version of the manuscript. The aims and novelty was emphasized more.

Reviewer 3 Report

In the presented work, the authors present a study of the effect of flavonoids isolated from the aerial parts of Hottonia palustris L. and Scleranthus perennis L. on lipid membranes. The authors carried out a complicated experimental study using FTIR spectroscopy and NMR technique. For the first time, these flavonoids were studied for their interaction with membranes and localization within membranes, as well as their influence on the dynamic and structural properties of model lipids. These data showed that the examined polyphenols were incorporated into the membrane region containing the polar head group of DPPC phospholipids at both 25°C and 45°C. At lower temperature a slight effect on the spectral region attributed to the ester carbonyl group was observed. In contrast, at 45°C, both compounds gave rise to slight effects in the spectral regions attributed to antisymmetric and symmetric stretching vibrations of CH2 and CH3 moieties. This is an interesting effect of temperature on the localization of interactions of flavones with lipid membranes. This effect may partly determine their biological activity and is critical for the usability of these compounds. This work is important for determining the affinity of natural compounds for lipid membranes. I recommend to accept this work after minor corrections. I recommend to show representative infrared spectra at 25 and 45 oC on one figure to show influence of temperature and flavonoids (on the upper panels) and several difference spectra below.

Minor editing of English language is required.

Author Response

Reviewer 3  comment:

Thank you very much for  very sophisticated and nice  general comment and among others   this specific one.

I recommend to accept this work after minor corrections. I recommend to show representative infrared spectra at 25 and 45 oC on one figure to show influence of temperature and flavonoids (on the upper panels) and several difference spectra below.

Our main goal was to established mechanisms of action for unfamiliar flavonoids  isolated from the aerial parts of H. palustris and S. perennis and examined their interaction and localization within lipid membranes. Intentionally,    we have  run FTIR measurements in two temperatures, namely 25 oC and 45 oC, in order to examine structural effects in two essentially different thermotropic phases of lipid bilayer membranes formed with DPPC: Lβ (below the phase pretransition) and Lα above the main phase transition temperature. In such a way we have access to information regarding the ordered and fluid phases. Hence we purposely, in order not to overexpress FTIR part of our study, limited the number of displayed spectra. But  at the same time, because these polyphenols are described for the first time, in our opinion it is worthy to show different spectral zones of lipids in various  temperatures because the tested compounds exhibited very interesting effect of temperature on the localization of flavones within lipid membranes. For the readers interested in the understanding the interaction of natural compounds at the molecular level it seems to be important and crucial. Such arguments  persuaded us to maintain figures  in the previous forms.